



# Forestlines in Italian mountains are shifting upward: detection and monitoring using satellite time-series

Lorena Baglioni[1], Donato Morresi[2], Matteo Garbarino[3], Carlo Urbinati[1], Emanuele Lingua[4], Raffaella Marzano[3] and Alessandro Vitali[1]

[1]Department of Agricultural, Food and Environmental Sciences, Marche Polytechnic University, Ancona, Italy

[2]Department of Forest Resource Management, Swedish University of Agricultural Sciences, Umeå, Sweden

[3]Department of Agricultural, Forest and Food Sciences, University of Torino, Torino, Italy

[4]Department of Land, Environment, Agriculture and Forestry, University of Padova, Padova, Italy

*Correspondence to*: Lorena Baglioni (*lorena.baglioni@pm.univpm.it*)

**Abstract.** The growing interest on the ecological effects of global warming and land use changes on vegetation, along with development of remote sensing techniques, fostered applied research on the successional dynamics at the upper limits of forests. The aims of this study were: i) to develop an automated methodology for mapping the current position of the uppermost italian forestlines; ii) to identify hotspots of change by the analysis of long-term greenness and wetness spectral dynamics. We carried on a Landsat-based trend analysis in buffer zones along the forestlines, testing differences between sparse and dense canopy cover classes and at different elevations and distances to the forestline. We used regional scale datasets and avoided to fix a minimum elevation threshold, in order to make the method replicable at different mountain ranges. For the spectral dynamics analyses, we used Landsat time-series of common vegetation indices for the period 1984–2023 and tested the significance of their long-term spectral trends with the Contextual Mann-Kendall test for monotonicity. We assessed that the highest forestlines are at the western sector in the Alps, and at the central one in the Apennines. We observed a common increase of the forest cover mainly close to the forestline and at lower elevations. Comparing greenness and wetness indices trends with the current canopy cover, the highest values were respectively in the sparse tree cover class, and in the dense one, particularly in the Alps.

**Keywords**: Landsat, remote sensing, Contextual Mann Kendall, treeline, spectral vegetation index.



## 1.    Introduction

A treeline is the contiguous ecotone along an altitudinal or latitudinal gradient  (Körner, 2008; Berdanier, 2010; Harsch et al., 2011).

Nowadays, treeline studies concern spatio-temporal dynamics to climate and/or land use changes (Malanson et al., 2011). Current treeline elevation and its spatial patterns derive from air temperature increase and past human activities that have modified treeline physiognomy and dynamics over time (Holtmeier et al., 2005; Harsch et al., 2011). Most European mountain landscapes have been shaped by humans since prehistoric times  through fire, deforestation and intensive grazing (Malanson et al., 2011; Vitali et al, 2018; Garbarino et al., 2020). In the Mediterranean region, the treeline elevation is threfore much lower than its potential climatic position (Körner 2012; Piper et al., 2016). Moreover, human acrivities have locally altered, directly or indirectly, species composition (Obojes et al., 2024) and induced new disturbances. For instance, in western Alps favouring *Larix decidua* Mill to *Pinus cembra* L. and promoting an invasive resprouter like *Alnus viridis* (Chaix) DC. (Dziomber et al., 2024, Motta et al, 2006).  In the Apennines planting *Pinus nigra* at high altitude facilitated its upward encroachment in treeline ecotones (Vitali et al., 2017). It is therefore reasonable to associate the upward advancement not only to global warming (Hansson et al., 2023) but also to successional dynamics (Ameztegui et al., 2016; Vitali et al., 2017) and to geomorphic processes (Leonelli et al., 2009). The ongoing development of remote sensing techniques and geographic information systems provided new opportunities in treeline studies (Holtmeier et al., 2020) such as detecting and monitoring the dynamic patterns of treeline shape and density (Fissore et al., 2015). Defining clear and easily replicable methods based on the application of modern technologies accessing to available datasets is fundamental for an accurate and large-scale treelines monitoring. At the local scale, aerial photography is commonly used (Ameztegui et al., 2016; Hansson et al., 2020; Nguyen et al., 2024) despite its limits due to image quality and availability (Morley et al., 2018). At larger scales, He et al. (2023) detected closed-loop mountain treelines integrating high resolution tree cover maps and  digital elevation models, whereas Wei et al. (2020) in the Western United States proposed an "alpine treeline ecotone detection index" (ATEI) based on the analysis of altitudinal and normalized difference vegetation index (NDVI) gradients. At the regional scale, supervised and unsupervised classification of multispectral images are widely used (Fissore et al., 2015, Chhetri and Thai, 2019), as well as detection techniques based on land cover maps combined with digital elevation models (e.g. Pecher et al., 2011).

Besides mapping, ongoing treelines research includes its spatio-temporal dynamics. Despite the coarser spatial resolution and the limitation of cloud cover, satellite optical imagery offers a finer time resolution by integrating data from several platforms and free processed time-series. It also provides spectral information for synchronic analysis of  vegetation changes (Gómez et al., 2016). After the open access to its database in 2008, the use of Landsat time-series has increased in recent decades (Zhu, 2017). Due to their spatial resolution (30 m) and temporal data availability, Landsat images have been largely used for vegetation dynamics monitoring, such as post-disturbance forest recovery (Morresi et al., 2019), greening or browning in different ecosystems (Bayle et al., 2024; Kumar et al., 2022; Rumpf et al., 2022) and to study alpine treelines  applying greening proxies like vegetation indices (Fissore et al., 2015; Tian et al., 2022). The NDVI is widely used in treeline dynamics, being



more sensitive in detecting biomass changes in open rather than in closed canopies (Choler et al., 2021; Wei et al., 2020; Arekhi et al., 2018; Bharti et al., 2012; Zou et al., 2022; Bayle et al., 2025)Fare clic o toccare qui per immettere il testo.. In the European Alps, Carlson et al. (2017) and Choler et al. (2021) used the annual maximum NDVI max values (NDVI-max) to analyse the  greening trends from Landsat and MODIS time-series. They assessed a significantly increasing spectral trend over the last two decades, mainly at north-facing slopes and in sparsely vegetated areas. Nevertheless, Bayle et al. (2024) remarked that the higher number of Landsat observations throughout the growing seasons can affect the NDVI-max trend analysis, causing false outcomes. It is true that annual NDVI-max increase with the number of available observations, and therfore their frequency must be taken into account. In the southwestern part of the European Alps, Bayle et al. (2025) studied greening trends using the annual max of kernel normalized difference vegetation index (kNDVI) (Camps-Valls et al., 2021), a nonlinear version of the NDVI, overcaming the overestimation of greening by the Harmonic Analysis of Time Series (HANTS), as reported by Choler et al. (2024). Bolton et al. (2018), instead, used the Enhanced Vegetation Index (EVI) for a Landsat based greening trend analysis of alpine treelines in the Canadian boreal zone. The EVI corrects the aerosol influence and canopy background noise and it is less affected by saturation than NDVI, being more sensitive to the NIR band (Huete et al. 2012). For this reason, it can detect the spectral behaviour of lower layers of vegetation while NDVI responds mainly to the RED band, which is involved in photosynthetic activity of the upper canopy layer. A single index can be combined with other vegetation indices to reduce the uncertainty on change detection analysis (Zhou et al., 2023; Schultz et al., 2016). EVI and NDVI can be considered grenness indices since they are linked to the photosintetic activity of vegetation by using NIR and RED bands, while wetness indices introduce short-wave infrared (SWIR) bands that are especially sensitive to water content of vegetation (Huete, 2012). Examples of wetness indices are the normalized difference moisture index (NDMI) and the normalized burn ratio index (NBR). This kind of indices are sensitive to shadowing, forest structure, leaf internal structure, vegetation moisture and density (Schroeder et al., 2011). In Landast-based forestry applications, indices derived from the Tasseled Cap Transformation (TCT) (Kauth and Thomas, 1976; Crist and Cicone, 1984) are also commonly used, since they are less affected by soil reflectance (Cohen and Goward, 2004). The Tasseled Cap Wetness index (TCW) considers the visible bands and both the SWIR1 and the SWIR2, and it is suitable to predict forest structural attributes, being slightly influenced by topographic variations, especially in closed conifer stands (Cohen et al., 1995). Another TCT index is the Tasseled Cap Angle (TCA) (Powell et al., 2010), combining the greenness and brightness information as defined in Crist and Cicone (1984) to assess the ratio between vegetated and non vegetated areas (Gómez et al., 2011). In our study we considered "forestline" the line separating the closed forest from the shrubland and grassland above, and "treeline" or "forestline ecotone" the surrounding area, the spatial pattern of which was not investigated due to the scale of analysis adopted. Forestlines ecotones are dynamic ecosystems and their monitoring on a regional scale can be carried on through the analysis of their spectral behaviour over time adopting different indices and tools. In this context, our study analyzed forestlines dynamics in the main Italian mountain ranges, the Alps and the Apennines. Our aims were

   1)   To define and monitor the position of the uppermost forestlines through an automated methodology;



2) To identify hotspots of change through satellite data, verifying whether and where forest recolonization dynamics are occurring.

In particular, we analysed the long-term greenness and wetness spectral changes of the uppermost forests and the contiguous forestline ecotones using Landsat-based trend analysis of time-series for the period 1984-2023, and we tested if greenness and wetness indices trends differed with elevation, forestline distance and canopy cover densities.

We hypothesised that greenness indices are more fitted for forest recolonization of open areas, while wetness ones for detecting gap-filling processes intercepting the spectral signal of lower leaf strata.


## 2.    Materials and methods

### 2.1.    Study area

The Alps and the Apennines are the two mountain ranges of the Italian peninsula. They extend respectively for 1300 and 1350 km: the Alps from West-to-East across northern Italy; the Apennines from NW to SE.. They also differ in climate, elevation range, and vegetation characteristics. In the Alps, annual precipitation range between 400 and 3000 mm, with rare summer dry

periods and cold winters (Isotta et al., 2014). Conifer forests prevail in the subalpine zone, where the main species are Norway spruce (*Picea abies* (L.) H.Karst.), European larch (*Larix decidua* Mill.) and/ or Swiss stone pine (*Pinus cembra* L.) (Fauquette et al., 2018). In the Apennines, the total annual precipitation range between 600 and 4500 mm (Vacchiano et al., 2017), with short and pronounced summer dry periods (Blasi et al., 2014).  Mixed broadleaf forests dominate at lower elevation, while

Common beech (*Fagus sylvatica* L.), locally mixed with silver fir (*Abies alba* Mill.), is the main species in the montane zone, except for rare locations in the central and southern parts of the Apennines, where also mountain pine (*Pinus mugo* Turra), European black pine (*Pinus nigra* J.F. Arnold), and Bosnian pine (*Pinus heldreichii* H.Christ) occur naturally. We selected the highest peak for each mountain group of the Alps and Athe Apennines, as defined by the Global Mountain Biodiversity Assessment (GMBA) inventory (Snethlage et al., 2022a, 2022b). We located the exact position of the peaks using the

nationwide Tinitaly Digital Elevation Model (DEM) v 1.1 (Tarquini et al., 2023). We then filtered the mountain groups and retained only the ones with highest peaks located on bare soil or in snow/ice covered areas, according to the Dynamic World land cover map (Brown et al., 2022). In this way we excluded also the mountain groups completely covered by forest or affetced by severe human impacts, such urban or built-up areas. In addition, for excluding the mountain peaks lacking the alpine bel thermoclimatic characteristics, we used the orotemperate, cryorotemperate and gelid thermotypes, derived from the

Bioclimates of Italy dataset (Pesaresi et al., 2017) based on the Worldwide Bioclimatic Classification System (WBCS) by Rivas-Martínez (1993). Others GMBA mountain groups have been removed after the previous selection based on land cover



and bioclimatic parameters, because of the Italian administrative and GMBA's boundaries limited the altitudinal range and forest distribution of some groups on the border in the following analyses (Sect. 2.2).

## 2.2. **Detection of forestlines**

We used the Tree Cover Density 2018 (TCD) of the European Environment Agency (EEA) derived from Sentinel-2 multispectral data as a reference for forest cover. TCD has a 10 m spatial resolution and provides information about the percentage of crown coverage in each pixel. According to the FAO "forest" definition (FAO, 2000), we selected only pixels having a TCD higher than 10 % to obtain a mask of forested areas. For each mountain group, we obtained the vertical distance

between each forest pixel and the DEM derived highest peak. We then selected the forest pixels with a vertical distance within the 1st percentile of all the distances and extracted the contours of the forestline by considering only the side of each selected forest pixel facing the mountain peak. We decided to not define a minimum altitudinal treshold to allow the method replicability on other mountain ranges of different elevations. Then, we joined polylines with linear gaps shorter than 30 m (corresponding to the Landsat spatial resolution). We considered only resulting polylines longer than 500 m to avoid highly

fragmented forestlines and to focus on more spatially extended and continuous ecotones and we removed closed loops to exclude the edges of forest gaps below the forestline. We outlined an area of interest around the uppermost forestlines where directing the analysis. Indeed, we defined a buffer zone of 250 m radius (Fig. 1c) along the lines to assess the presence of significant spectral changes in a gradient from closed forest to grassland. We also sampled points at 10 m distance along the detected forestlines to assess the mean, median and maximum elevation of all the detected forestlines, grouped by mountain

groups or mountain ranges. We processed the data in the R software environment (v. 4.3.2) using the "terra" (Hijmans, 2023), "callr" (Csárdi and Chang, 2024) and "future.apply" (Bengtsson, 2021) packages, and with QGIS software (v. 3.34.1).





**Figure 1 - (a) Selected peaks (triangles) along the Italian Alps (light blue) and Apennines (orange); (b) Detected forestlines (red polylines) on a ESRI satellite image (ArcGIS/World_Imagery) of Punta Ramière (3330 m a.s.l.) in the Montgenèvre Alps GMBA group; (c) a 3D graphical model based on the Tinitaly DEM and the TCD forest mask (TCD >10 %) used for the forestlines detection and the buffer area definition (yellow area).**

2.3.  **Trend analysis of vegetation indices**

Landsat data provide medium resolution images (30 m pixel size) from 1984 to 2023, that are commonly used in treeline studies (Arekhi et al., 2018, Bharti et al., 2012, Morley et al., 2019, Garbarino et al., 2024) since they offer a good compromise between space and time resolution (Hansson et al., 2020) at regional scale. We collected multispectral images from June 1st to September 30th of each year to analyse the spectral behaviour of forest vegetation during the growing season and to reduce the

effect of snow cover and alpine meadows drying up during the summer months.





In particular, we used Landsat Level-2 Collection 2 images acquired by the TM, ETM+, OLI and OLI-2 sensors. After masking pixels covered by snow, clouds, cloud shadows and water, we produced reflectance composites based on the medoid compositing approach (Flood 2013). We computed common vegetation indices from reflectance composites that we grouped into i) greenness indices: normalized difference vegetation index (NDVI) (Tucker, 1979), enhanced vegetation index (EVI)
(Huete et al., 2002) and tasseled cap angle index (TCA) (Powell et al., 2010); and ii) wetness indices: normalized burn ratio (NBR) (García et al., 1991), normalized moisture index (NDMI) (Gao, 1996) and tasseled cap wetness index (TCW) (Crist, 1985). Finally, we masked the 40 year-long time-series using the buffer areas around each selected forestline (Sect. 2.2).
We assessed the significance in the monotonicity of the spectral trends, i.e. strictly increasing or decreasing, derived from vegetation indices time-series by applying the non-parametric Contextual Mann Kendall (CMK) statistical test (Neeti et al.,
2011). The CMK test is an estimator of the monotonicity of trends, based on the Mann-Kendall (MK) test, which takes into account the trends in the neighbouring pixels within a 3 x 3 kernel. In this way, the spatial autocorrelation is considered, thus improving the detection of spatial patterns characterised by homogeneous spectral trends. Specifically, we used the *"ConMK"* R package (available at https://github.com/geoporttishare/ConMK). The TAU statistics produced by the MK test ranges between +1 and -1, with positive values indicating an increasing trend, while negative values are associated with decreasing
trends. Before checking the occurrence of significant trends by computing the p-value ($\alpha$) associated with the TAU statistics, we pre-processed time-series in two steps. First, we filled missing data at the pixel level through linear interpolation and considering also one-year gaps while discarding pixels with longer data gaps. Second, we removed the autocorrelation in the time-series by applying the pre-whitening procedure proposed by Wang and Swail (2001) and implemented in the *"ConMK"* R package.


## 2.4. **Assessment of the spectral trends**

For each group of vegetation indices, i.e. grenness and wetness, we selected only those pixels that exhibited a highly significant trend ($\alpha < 0.005$), as proposed in Choler et al. (2021) for all the indices (Fig. 2).





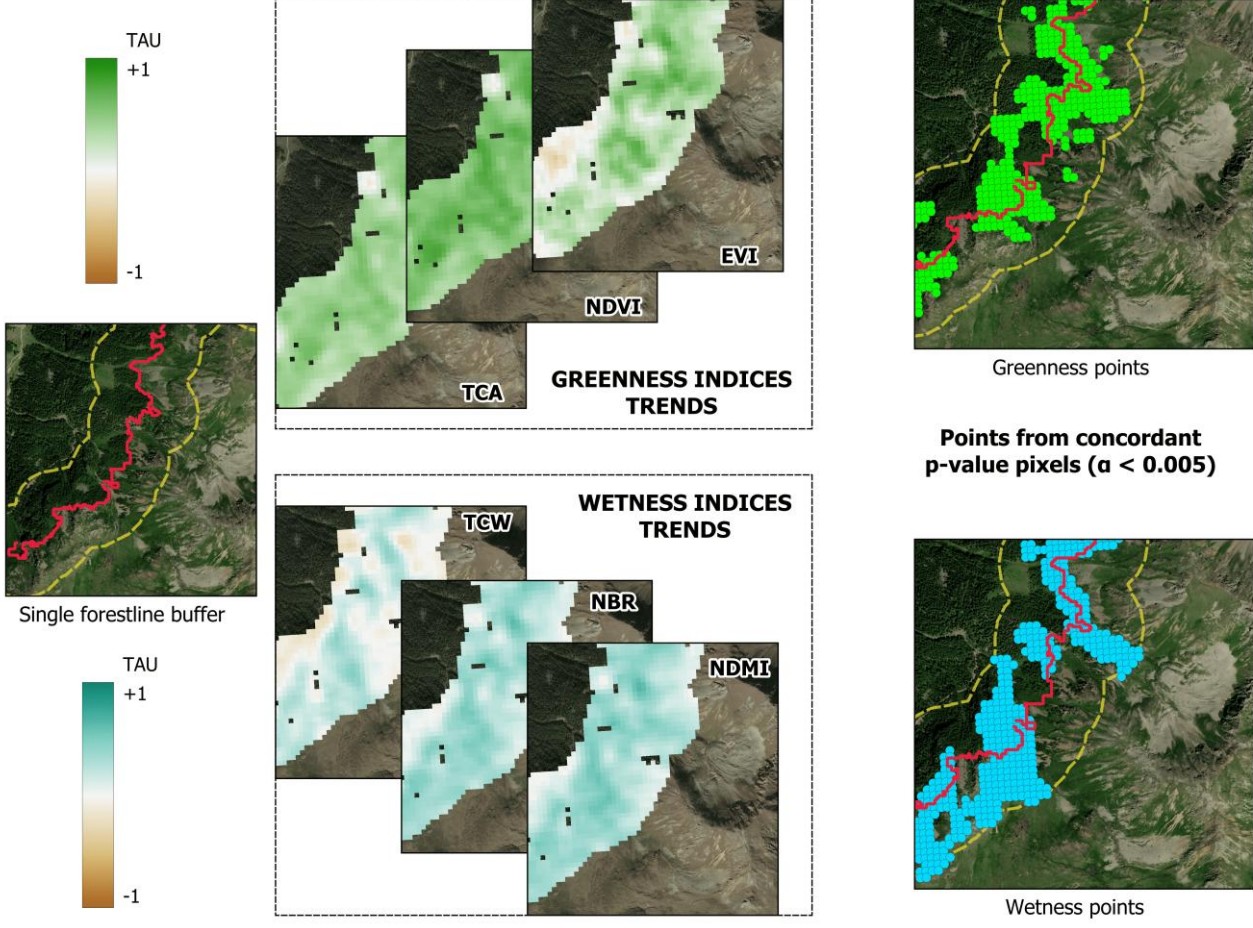

**Figure 2 - Example of the extraction of the highly significant (α < 0.005) greenness (top) and wetness (bottom) points trends in the buffer area (yellow dotted line) around a single forestline (full red polyline). Base map: ESRI Satellite (ArcGIS/World_Imagery).**

We used the points corresponding to the resulting pixel centroids to extract the mean TAU value among the vegetation indices

and the elevation from the Tinitaly DEM. The original TCD excludes shrublands, dwarf pine or green alder in alpine areas (Copernicus Land Monitoring Service, 2021) but we resampled it to 30 m by average and assigned to each pixel centroid also the mean tree canopy cover.

Then, we excluded points with negative trends (TAU < 0) and with TCD = 0 that corresponded to areas without a tree canopy cover (e.g. grasslands) and where the significant increasing spectral trend of the last 40 years was probably due to factors and

dynamics different from forest recolonization. We assigned the elevation value of the nearest forestline point to the resulting wetness and greenness points using the "join attributes by nearest" tool in QGIS. In this way, we obtained the Euclidean distance of each trend point to the forestline and the difference in elevation, which we used to classify the points into above or below the forestline. Specifically, we identified the relative position to the forestline of each points by multiplying the





Euclidean distances by the sign of the difference in elevation. We carried out the analysis by considering the Alps and

Apennines separately, due to their altitudinal, climatic and vegetation differences, as previously described (Sect. 2.1). For each

mountain range, we obtained two sets of 40.000 randomly sampled points, broadly containing a equal number of greenness

and wetness significant trends. We then grouped the sampled points into three tree canopy cover categories according to TCD:

i) sparse canopy cover (TCD <10 %); ii) moderate-to-dense canopy cover (10 % < TCD < 80 %); iii) dense canopy cover

(TCD > 80 %). We then assessed the relationship between TAU values and the canopy cover, the elevation and the distance

to forestline, taking into account the mean values of each segment of forestline. We used a Wilcoxon test (Wilcoxon, 1945) to

verify significant differences between the mean TAU among wetness and greenness indices averaging the mean TAU values

of each canopy cover class in each forestline buffer. Finally, we built Generalized Additive Models (GAM) (Hastie and

Tibshirani, 1990) using the cubic spline smoother of the "mgcv" R package (Wood, 2011) to test the presence of a significant

non-linear relationship between the mean TAU values and i) the elevation and ii) the distance to forestline, without considering

the TCD classes.




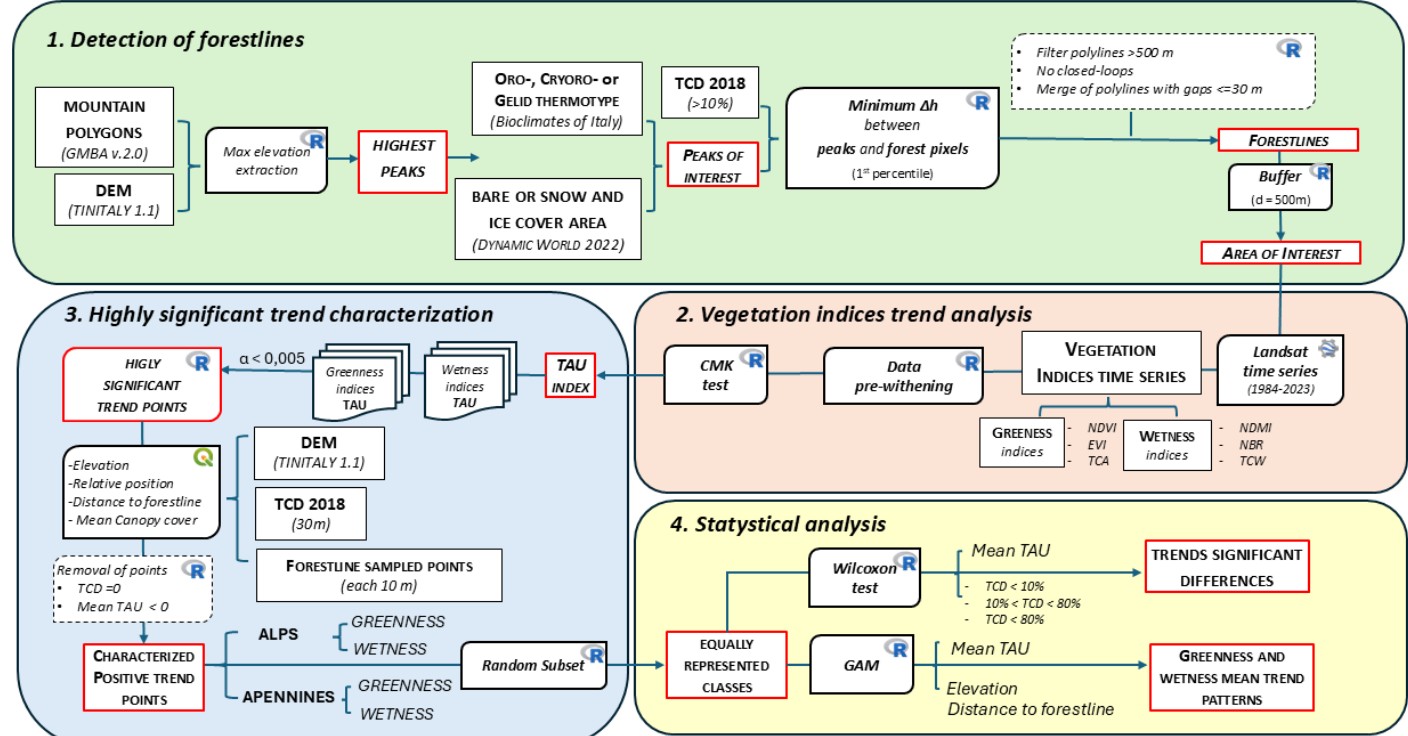

**Figure 3 – Flow chart of the analytical process: input data (black rectangles), outputs (red rectangles) and analyses (irregular rectangles). Abbreviations: DEM (digital elevation model), TCD (tree cover density), NDVI (normalized difference vegetation index), EVI (enhanced vegetation index), TCA (tasseled cap angle index), NDMI (normalized difference moisture index), NBR (normalized burn ratio), TCW (tasseled cap wetness index), CMK (contextual mann-kendall test), GAM (generalized additive model).**

## 3.    Results

### 3.1.    Forestlines extraction

We identified and processed 60 mountain peaks, 44 in the Alps and 16 in the Apennines (Table A1-Appendix A). We obtained approximately 5760 km of forestlines with a mean elevation of 2088 ± 193 m a.s.l. in the Alps and 1758 ± 161 m a.s.l. in the Apennines, with a maximum elevation respectively of 2500 and 2383 m a.s.l. In the Alps, the lowest forestline elevations were in the prealpine groups due to the lack of a suitable altitudinal gradient, whereas the highest ones were mainly in the western sector (Fig. 4a). In the Apennines, the lowest and less extended forestlines, if compared to the forested area, were in the northern sector, while the highest ones were in Central Italy: the Majella (MJ), Sirente Velino (SVE) and Marsicani (MM) mountain groups (Fig. 4b). The total area of interest obtained was of approximately 1880 km$^2$ .





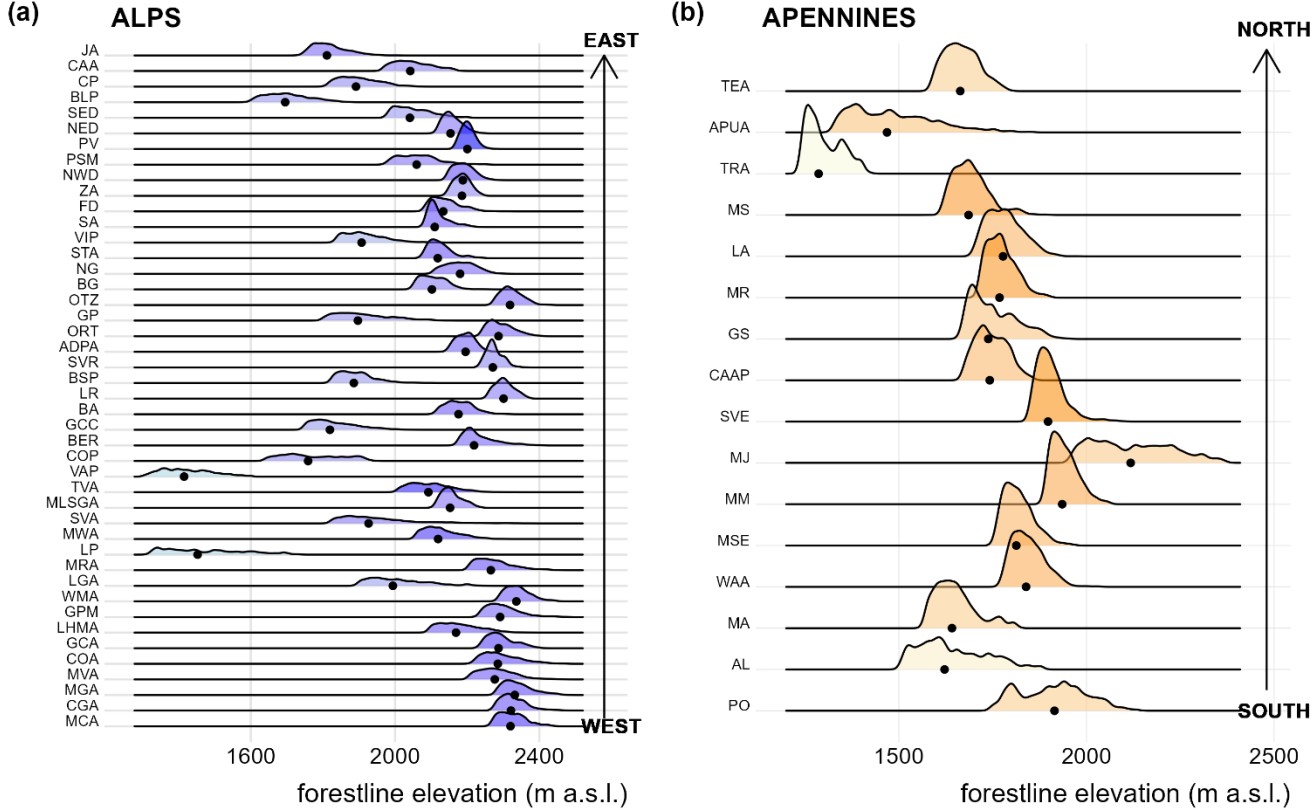


**Figure 4 - (a) Forestline elevation ranges of the mountain groups in the Alps (n = 44), sorted by longitude (West – East); (b) Forestline elevation ranges of the mountain groups of the Apennines (n = 16), sorted by latitude (South – North). Black dots is the median value of each interval5; colour intensity of ridges increases with forestline length and forested area ratio. Mountain groups codes are available in Table A1 (Appendix A).**


## 3.2. Trend analysis and performances

We obtained 28.81 % of highly significant (α < 0.005) trends pixels for greenness and 19.69 % for wetness, considering only pixels with concordant p-values on all of the forming indices. The majority of TAU values were positive (Fig. 5) at both index types, with respectively 97.8 % and 99.8 % in the Alps, and 96.3 % and 99.7 % in the Apennines.






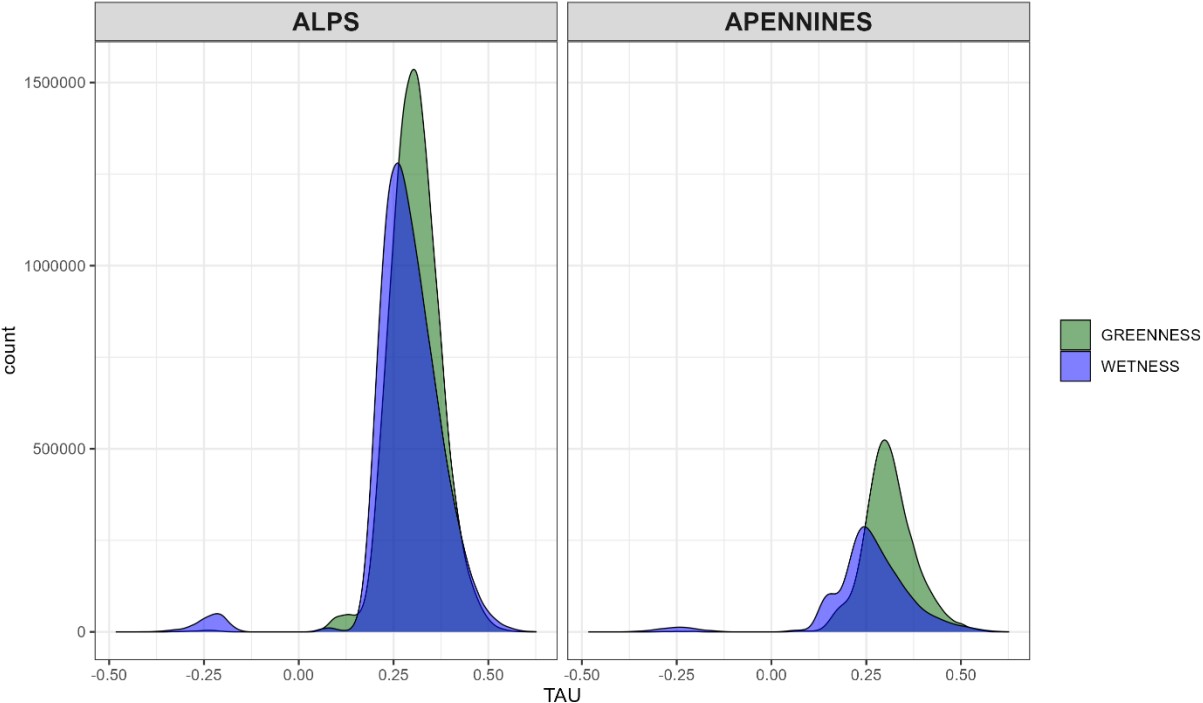

**Figure 5 – Distribution of the highly significant (α < 0.005) wetness (blue) and greenness (green) pixels frequency with different TAU values).**

In total, we obtained  242233 greenness and 221554 wetness highly significant positive trend pixels for the Alps,  and 76348 and 508745 rispectively for the Apennines. With sparse and moderate-to-dense canopy cover, the greenness and wetness positive trends were mainly near the forestline in both mountain ranges, and decreased in both directions, but mainly upwards where tree covered areas are gradually replaced by high-altitude grasslands. With dense canopy cover, only wetness positive trends in the Apennines showed a similar distribution, differently from greenness trends and from both types in the Alps, where

the highest concentration was below and distant from the forestline (Fig. 6).
.





**Figure 6 – Positive wetness (blue) and greenness (green) trend pixels density according to their distance to the forestline (black dashed line) in different canopy cover classes: sparse (TCD < 10 %), moderate-to-dense (10 % < TCD < 80 %) and dense (TCD > 80%). Negative and positive values represent distances below and above the forestline, respectively.**

Significant differences between TAU trends and canopy cover classes occurred mainly in the Alps (Fig. 7). The highest ones were for the wetness indices in the dense canopy cover. For greenness, sparse canopy cover class had higher mean TAU than moderate-to-dense and  dense classes. In the Apennines only greenness values highlighted a significant difference between the sparse and the dense canopy cover class.




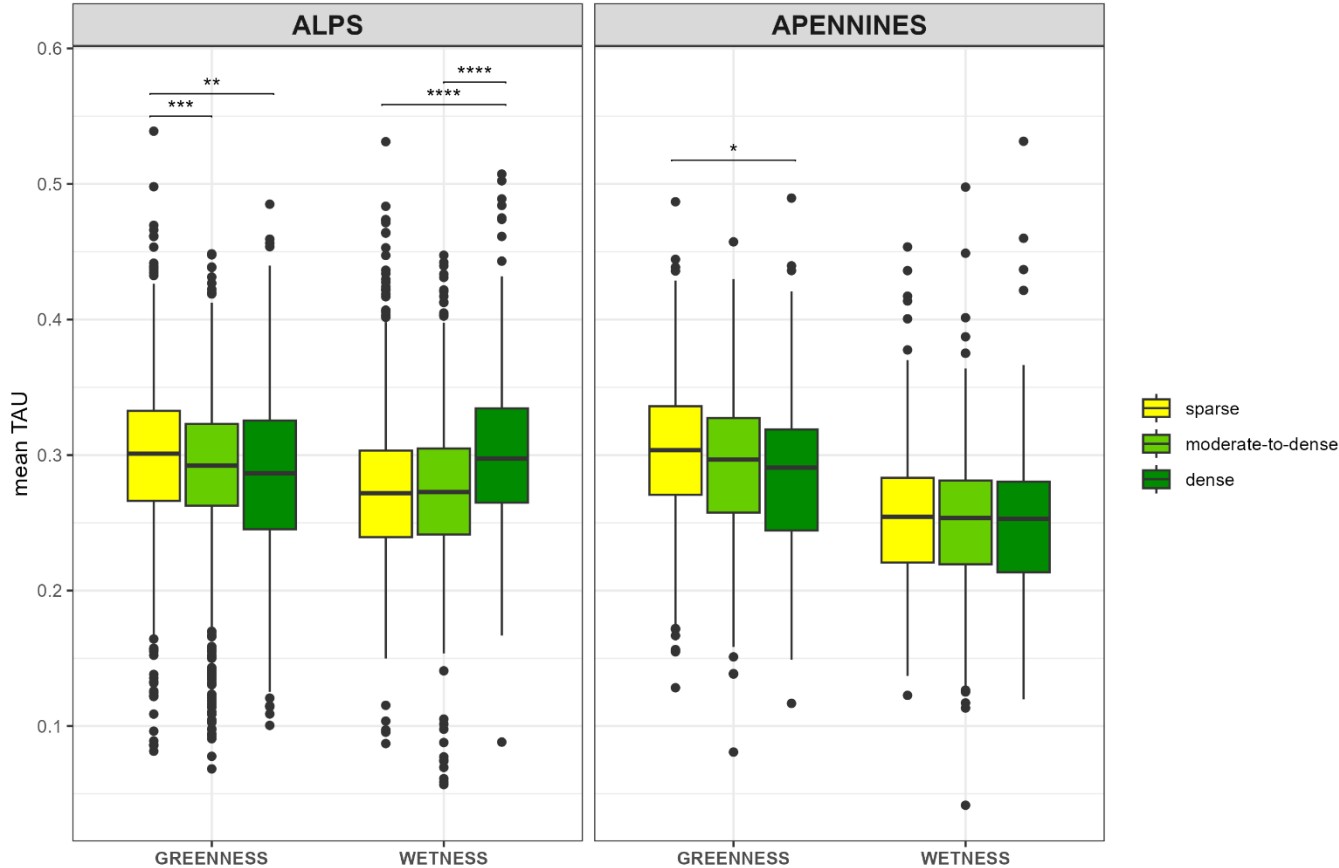

**Figure 7 – Boxplots of the mean TAU values of wetness and greenness trends in the Alps (left panel) and in the Apennines (right panel). The mean values in each forestline buffer account for three different canopy cover classes: sparse (yellow), moderate-to-dense (green), dense (dark green). Significant differences of Wilcoxon test are indicated with: * ($\alpha <= 0.05$), ** ($\alpha <= 0.01$), *** ($\alpha <= 0.001$), **** ($\alpha <= 0.0001$). For more information on the canopy cover classes percentage in the Alps and in the Apennines refer to Fig. B1 (Appendix B).**

GAM models did not detect a statistically significant relationship between grenness/wetness mean TAU values, the distance to forestline, and the elevation (Fig. 8). Relatively similar patterns of the two indices mean trends appeared at both mountain ranges but with a higher varibility in the Apennines. In general, TAU greenness values were higher than wetness ones. In the Alps, greenness increased moving upwards to the forestline with a first culmination close to and below it, followed by a decrease and subsequent increase, with the highest mean values over 200 m. In the Apennines instead, the mean TAU values increased close to the forestline with a culmination above it (about 100 m distance), followed by a decrease, with the lowest values above 200 m. In both the mountain ranges we observed a decrease of the greenness trends from lower elevations up to about 1750 m a.s.l. in the Alps and 1500 m a.s.l. in the Apennines (Fig. 8b). Thereafter, the mean TAU values increase



275    progressively in the Alps up to 2300 m a.s.l., but decrease slightly in the Apennines to around 1700 m a.s.l. to rise again up to
the altitudinal limit.

Wetness trend related to the forestline distance is very flat in the Alps and relatively similar to that of the greenness, whereas
in the Apennines, the trend is far more variable and increasing progressivley from forestline uo ti 200 m above it (Fig. 8a).
Wetness curves decrease for both Alps and Apennines from the lower elevations to about 1600 m a.s.l (Fig. 8b), with a more
280    pronounced slope for the Apennines. Then they both rise up to about  2100 m a.s.l. with a steeper and fluctuating trend again
in the Apennines.

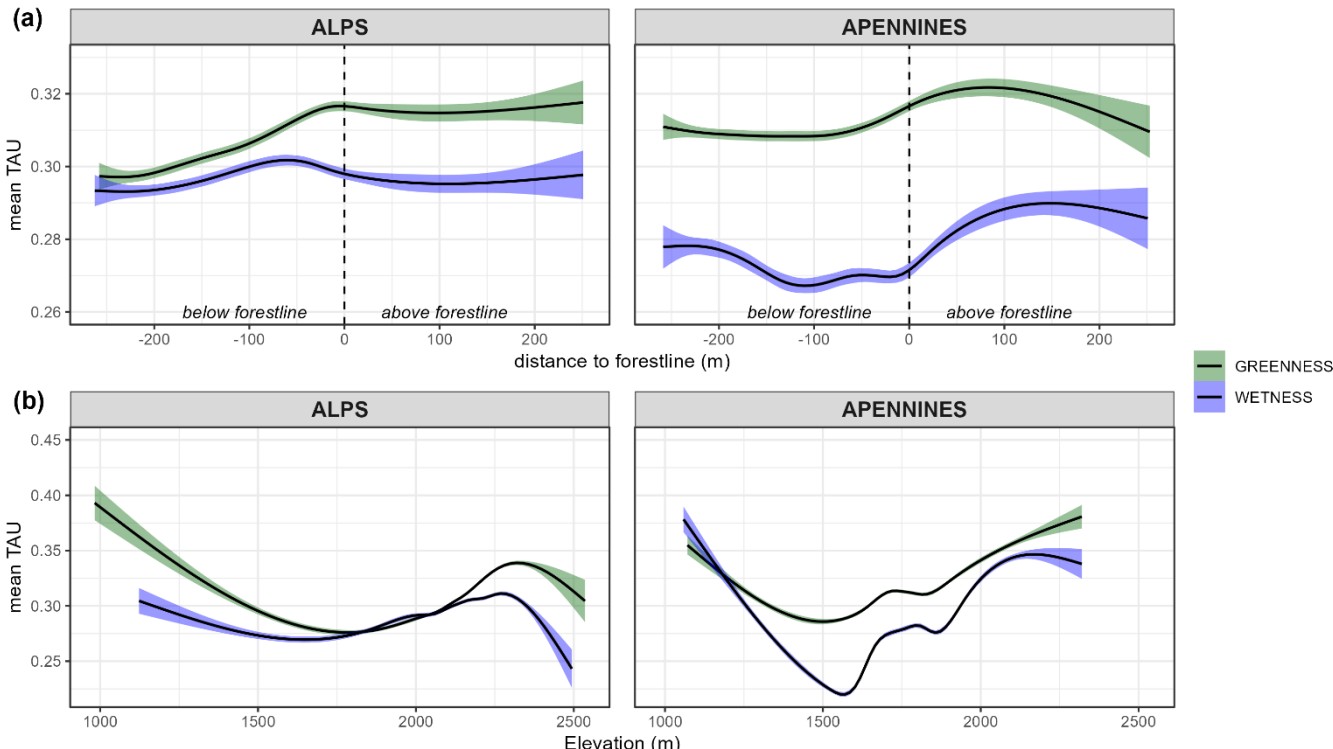

285    **Figure 8 - Wetness (blue) and greenness (green) GAM functions with a level of confidence of 0.95, according to the elevation (a) and
to the distance to the forestline (b) in the Alps (left) and in the Apennines (right). We considered the mean values of each forestline
buffer. The models are not statistically significant (α > 0.05).**





## 4. Discussions

### 290 4.1. The uppermost forestlines detection method

The proposed forestline detection method is applicable at different spatial scales and in different geographic regions as it does not establish elevation thresholds and it can be based on regional datasets or other existing digital elevation models and forest masks. In addition, the method is also not exclusively based on climatic parameters, therefore applicable for the detection of human impacted forestlines. We considered as forestlines those closer to their potential climatic limit (e.g. tree species line), 295 excluding forest margins at lower elevations and highly fragmented forestlines. We based the detection on the smallest elevation differences between the highest peaks and the forested pixels resulting from recent satellite derived data (TCD 2018). According to some authors, Italian and many others northern emisphere forestlines are not "climatic" treelines due to severe human limitations (e.g. grazing, fire and deforestation) that altered altitudinal position, spatial pattern and tree composition (Motta et al., 2006; Malanson et al., 2011; Piermattei et al., 2016; Vitali et al., 2018; Holtmeier and Broll, 2020). In Italian 300 mountains, forest upward expansion has been favoured mainly by past large-scale disturbances, and took place mainly at warmer aspects (Malandra et al., 2019). Recurrent human direct impacts on these ecotones since the Holocene times have greatly affected vegetation structure and composition (Foster et al., 1998), and recent silvo-pastoral abandonment at high elevation sites triggered secondary succession (Debussche et al., 1999). The different forest cover and the bioclimatic features of the selected mountain groups provide a representative sample of the forestline trends along the Italian peninsula. The mean 305 forestlines elevations detected, confirm previous studies in the Alps (Caccianiga et al., 2008; Lingua et al., 2008; Diàz-Varela et al., 2010; Gilles et al., 2023) and in the Apennines (Vitali et al., 2018; Bonanomi et al., 2020). However, since the proposed method is closely dependent on the available regional/national datasets, some exclusion occurred with mountain groups at transnational borders.

### 310 4.2. Long-term greenness and wetness spectral trends

Overall, greenness and wetness raising trends were recorded at both mountain ranges in line with the ongoing natural reforestaion processes (Vitali et al., 2018; Garbarino et al., 2020; Anselmetto et al., 2022). Pixel density distribution of Alps and Apennines are globally very similar, but some differences appear for the dense canopy cover class in the Apennines, where the wetness indeces have the highest peak just below the forestline. This could be attributed to the different species composition 315 at the two mountain ranges. Along the Apennines, with the exception of some scattered locations with *Pinus* spp*., Fagus sylvatica* is practically the only upper forestline species (Piermattei et al., 2014; Vitali et al., 2017) . This would confirm the long term impact of human activity (Körner, 2012) and explain the occurrence of "abrupt" (Harsch et al., 2009) and static treelines (Bader et al., 2021; Bonanomi et al., 2018) given the very limited seed dispersal efficiency of beech (Vitali et al., 2017). Dominant species colonization rates, reproduction, seed-dispersal strategy and vitality of the occurring species are





relevant issues when comparing forestlines shifts in Alps and Apennines (Holtmeier, 2009; Compostella et al., 2017; Garbarino et al., 2020). In general gap-filling processes prevail at the deciduous Apennines forestlines (Malandra et al., 2019; Vitali et al., 2018), whereas in the Alps the coniferous species treeline (e.g. *Larix decidua*, *Pinus cembra* and *Picea abies*) foster tree encroachment at higher elevations. The beech regeneration (both by seeds and suckers) in the Apennines abrupt treelines have favoured processes of canopy thickening and gap filling below and near the forestline, better intercepted by wetness indices

that are more sensitive to spectral response of the less exposed vegetation. Wetness indices are particularly sensitive to water content in both soil and plant especially in canopy leaf tissues. For this reason, we believe that significant increasing trends in areas with a dense canopy cover could be associated to crown thinning and biomass increase, as in gap filling, while in areas with sparse canopy cover, to new encroachment in open areas.

         Considering the magnitude of changes rather than their frequency, in the Apennines we assessed a significant difference only

for greenness mean TAU that resulted lower in dense than in sparse canopy cover conditions. We assume that the drier climatic conditions of the Apennines, may have influenced the positive trends of wetness indices, reducing the TAU variability in different canopy classes. The Wilcoxon test revealed the most significant variations in the Alps, where summer drought is not a limiting factor as in the Apennines. This hypothesis is confirmed by the higher mean TAU values of greenness in the sparse canopy class, whereas those of wetness refer to the dense one. Carlson et al. (2017) in the French Alps found a stronger

greening signal in low-shrubs and open areas (e.g. grasslands or rocky habitats) than in forested areas. As well McManus et al. (2012) in the forest-tundra ecotones in Canada found higher greening in shrub and grass canopy classes. Sometimes, the greening of sparse open areas may be affected by melting glaciers (Rumpf et al., 2022), inducing a possible increase in soil moisture and influencing wetness trends too. Without considering the canopy cover class, the most relevant changes in the Apennines occurred above the forestline and at the lowest (< 1500 m a.s.l.) and highest (> 2000 m a.s.l.) elevations. Adding

information about the forest structure could help identifing if the spectral signal sourced mostly from shrubs or newly established trees. In the Apennines, the highest mean TAU values above the forestline can be due to species like *Juniperus communis* L., *Pinus mugo* Turra and *Vaccinum mirtillus* L., that facilitate the upward migration of beech trees (Bonanomi et al., 2021). In the Alps, we found a steadier increase of TAU greenness and wetness from below to above the forestline. This confirms that diffuse treelines are more common in the Alps (Garbarino et al., 2020). Some authors used LiDAR data from the

Global Ecosystem Dynamics Investigation (GEDI), integrated with Landast and Sentinel-2 data (Potapov et al., 2021; Tolan et al., 2024; Lang et al., 2023) to assess canopy height, vertical canopy structure and surface elevation, with the aim of monitoring forest ecosystems and carbon fluxes. This approach could be adopted in monitoring ecotones like treelines (Bolton et al., 2018), also to  predict future vegetation scenarios and provide suitable management options (Morales-Molino et al., 2022).

Considering the greeneess and wetness mean values of TAU trends only as a function of elevation, without the forestline distance information, their higher values are mainly at lower sites, where temperature is less limiting and the past human impact was greater (Malandra et al., 2019; Anselmetto et al., 2022). Furthermore forests at lower elevations are more accessible and usually most intensively managed in the past, but now are extensively abandoned (Malandra et al., 2019; Garbarino et al.,



2020). Above the mean forestline elevation of both Alps and Apennines, the mean TAU values increase and then decrease at
higher altitudes where the number of pixels with significant increasing trends is also less. In the Apennines, a second short but
clear decrease above 1750 m a.s.l. may depend on the frequent abrupt beech treelines where the forest margin is sharply
separated from areas with sparse and different vegetation. This common trend for both mountain ranges confirms an upward
recolonisation process at the Italian anthopogenic treelines. The decreasing magnitude observed at higher elevation with the
increasing distance from the forestline is probably due to the larger distance from seed trees (Vitali et al., 2018), the more
limiting effect of temperature and the synergic effect of topography and microclimate.

Uncertainty remains about causes of the spatial variability of trends, as noted also by Choler et al. (2021). The time-series span
and the spatial resolution of satellite images are crucial aspects in the definition of this kind of ecological models. Nevertheless,
the integration of different data sources (e.g. LiDAR) and the modelling of further environmental, climate and anthropogenic
drivers can be essential for a better understanding of the current and future forestline dynamics.

**5.    Conclusions**

This work introduced a new method to demarcate the upper forestlines at a regional scale, in environments where they do not
match the climatic boundary (i.e. the 6 °C isotherm sensu Körner and Paulsen, 2004). We proposed several parameters to
define only the forestlines closest to their potential position, detecting the ones nearest to mountain top with land cover and
bioclimatic features. The use of TCD, with national or European digital terrain models, makes it applicable in most parts of
Europe, but with similar datasets also in other regions of the world and at different scale of analysis. High spatial resolution,
wide geographical coverage and open data availability policies are important factors for the replicability of the algorithm and
for ensuring the quality of both the detection results and the trend analysis. Landsat images permit to analyse 40 year-long
time-series with a suitable spatial resolution. Even though the two types of indices have different targets (greeneess indices for
photosynthetic activity and wetness indices for water content), the results were congruous and emphasized the altitudinal
expansion of the forestline ecotone at national scale. Wetness indices were more sensitive in areas with denser canopy cover,
probably due to gap-filling processes and increasing biomass. Greenness indices detected more relevant trends, especially in
areas with sparse or medium canopy cover, probably where recent tree encroachment occurred in previously open areas.

In the current context of climate change and post-abandonment dynamics, the implementation of semi-automatic methods for
detection and monitoring of vegetation spatial patterns and modelling of spectral trends is an added value. With this study, we
defined hotspots of changes through different spectral indices and can be the basis for future landscape-scale analyses to better
assess the relationships between climate, topography, vegetation dynamics and forest structure changes..

## 6. Appendices

### Appendix A:

**Table A1 - Selected Italian GMBA mountain groups, whose peaks having land cover and thermotypes suitable for the proposed algorithm. Elevation information was extracted from the Tinitaly DEM, limiting the areas to the national administrative boundaries.**

| ID | | Mountain group statistics | | | | Forestline statistics | | | | | | |
|---|---|---|---|---|---|---|---|---|---|---|---|---|
| Code | Name | highest peak (m) | min elevation (m) | forested area ($km^2$) | area ($km^2$) | length (km) | mean length (km) | n° | mean elevation (m) | median elevation (m) | max elevation (m) | Length/forested area (km $km^{-2}$) |
| **ALPS** | | | | | | | | | | | | |
| **ADPA** | Adamello-Presanella Alps | 3552 | 250 | 657 | 1379 | 115 | 0.98 ± 0.56 | 118 | 2196 ± 27 | 2196 | 2343 | 0.175 |
| **BA** | Bergamasque Alps | 3030 | 198 | 815 | 1417 | 151 | 1.06 ± 0.74 | 146 | 2179 ± 39 | 2176 | 2332 | 0.185 |
| **BER** | Bernina Range | 3998 | 0 | 431 | 976 | 76 | 1.13 ± 0.78 | 67 | 2233 ± 46 | 2219 | 2416 | 0.176 |
| **BG** | Brenta group | 3163 | 190 | 455 | 706 | 68 | 1.19 ± 0.97 | 57 | 2107 ± 37 | 2102 | 2272 | 0.149 |
| **BLP** | Bellunese Prealps | 2457 | 26 | 1058 | 1494 | 128 | 1.76 ± 2.62 | 73 | 1701 ± 60 | 1695 | 1889 | 0.121 |
| **BSP** | Brescia Prealps | 2250 | 125 | 811 | 1124 | 96 | 2.16 ± 1.59 | 44 | 1892 ± 48 | 1886 | 2109 | 0.118 |
| **CAA** | Carnic Alps | 2778 | 241 | 1202 | 1720 | 173 | 1.36 ± 1.38 | 92 | 2049 ± 51 | 2042 | 2193 | 0.144 |
| **CGA** | Central Graian Alps | 3747 | 660 | 136 | 619 | 26 | 1.09 ± 0.6 | 24 | 2329 ± 37 | 2321 | 2479 | 0.191 |
| **COA** | Cottian Alps | 3286 | 0 | 345 | 823 | 69 | 1.08 ± 0.67 | 64 | 2295 ± 51 | 2285 | 2481 | 0.2 |
| **COP** | Como Prealps | 2242 | 0 | 569 | 847 | 78 | 3.12 ± 3.62 | 25 | 1773 ± 82 | 1759 | 1950 | 0.137 |
| **CP** | Carnic Prealps | 2703 | 132 | 933 | 1304 | 125 | 2.21 ± 3.1 | 78 | 1899 ± 55 | 1892 | 2092 | 0.134 |
| **FD** | Fiemme Dolomites | 2844 | 186 | 1445 | 1971 | 219 | 1.56 ± 1.4 | 141 | 2140 ± 40 | 2134 | 2326 | 0.151 |
| **GCA** | Grand Combin Alps | 3725 | 555 | 141 | 538 | 29 | 1.14 ± 0.8 | 82 | 2293 ± 38 | 2287 | 2420 | 0.206 |
| **GCC** | Gruppo Camino-Concarena | 2547 | 158 | 1373 | 2003 | 164 | 1.51 ± 1.62 | 106 | 1833 ± 63 | 1819 | 2126 | 0.119 |
| **GP** | Garda Prealps | 2251 | 12 | 1273 | 1686 | 168 | 1.87 ± 1.72 | 90 | 1911 ± 76 | 1897 | 2163 | 0.132 |
| **GPM** | Grand Paradis Massif | 4060 | 236 | 537 | 1563 | 93 | 1.01 ± 0.61 | 29 | 2299 ± 45 | 2291 | 2491 | 0.174 |
| **JA** | Julian Alps | 2751 | 311 | 286 | 439 | 48 | 1.17 ± 0.64 | 41 | 1819 ± 47 | 1811 | 2017 | 0.168 |
| **LGA** | Ligurian Alps | 2650 | 0 | 1503 | 2051 | 169 | 1.88 ± 1.58 | 90 | 2008 ± 78 | 1994 | 2262 | 0.112 |





| | | | | | | | | | | | |
|---|---|---|---|---|---|---|---|---|---|---|---|
| LHMA | Lanzo and Haute Maurienne Alps | 3676 | 0 | 611 | 1291 | 114 | 1.5 ± 1.5 | 76 | 2177 ± 56 | 2169 | 2387 | 0.186 |
| LP | Ligurian Prealps | 1743 | 0 | 875 | 1043 | 50 | 2.65 ± 5.04 | 19 | 1474 ± 109 | 1452 | 1743 | 0.057 |
| LR | Livigno Range | 3436 | 0 | 180 | 630 | 28 | 1.01 ± 0.64 | 26 | 2303 ± 26 | 2301 | 2376 | 0.153 |
| MCA | Mont Cenis Alps | 3468 | 515 | 126 | 352 | 27 | 0.84 ± 0.6 | 26 | 2325 ± 37 | 2320 | 2444 | 0.216 |
| MGA | Montgenevre Alps | 3301 | 277 | 797 | 1432 | 159 | 1.31 ± 1.21 | 121 | 2339 ± 41 | 2332 | 2500 | 0.2 |
| MLSGA | Mont Leone and Saint Gothard Alps | 3551 | 0 | 132 | 379 | 21 | 1.51 ± 1.13 | 18 | 2157 ± 28 | 2153 | 2245 | 0.159 |
| MRA | Monte Rosa Alps | 4607 | 197 | 584 | 1382 | 113 | 1.75 ± 1.38 | 110 | 2275 ± 51 | 2266 | 2469 | 0.194 |
| MVA | Monte Viso Alps | 3841 | 275 | 1041 | 1971 | 192 | 1.32 ± 1.28 | 86 | 2282 ± 50 | 2277 | 2486 | 0.185 |
| MWA | Mischabel and Weissmies Alps | 3610 | 223 | 194 | 392 | 42 | 1.02 ± 0.58 | 41 | 2127 ± 48 | 2119 | 2313 | 0.216 |
| NED | Northeastern Dolomites | 3261 | 529 | 744 | 1474 | 118 | 1.11 ± 0.72 | 108 | 2158 ± 26 | 2154 | 2253 | 0.159 |
| NG | Nonsberg Group | 2953 | 200 | 693 | 966 | 119 | 1.61 ± 1.33 | 72 | 2181 ± 41 | 2180 | 2319 | 0.172 |
| NWD | Northwest Dolomites | 3343 | 276 | 803 | 1426 | 156 | 1.11 ± 0.74 | 141 | 2190 ± 29 | 2188 | 2298 | 0.194 |
| ORT | Ortler Alps | 3892 | 270 | 715 | 1768 | 123 | 1.28 ± 0.9 | 99 | 2293 ± 36 | 2287 | 2419 | 0.172 |
| OTZ | Ötztal Alps | 3723 | 0 | 300 | 1024 | 51 | 0.97 ± 0.61 | 53 | 2323 ± 30 | 2319 | 2438 | 0.171 |
| PSM | Pale di San Martino | 3190 | 260 | 456 | 742 | 74 | 1.8 ± 2.05 | 40 | 2064 ± 55 | 2060 | 2263 | 0.161 |
| PV | Puster Valley | 3424 | 807 | 257 | 531 | 61 | 1.08 ± 0.7 | 56 | 2202 ± 19 | 2200 | 2288 | 0.236 |
| SA | Sarntal Alps | 2773 | 239 | 654 | 1115 | 124 | 1.28 ± 0.86 | 97 | 2119 ± 29 | 2110 | 2245 | 0.19 |
| SED | Southeast Dolomites | 3217 | 351 | 446 | 659 | 54 | 1.52 ± 1.09 | 34 | 2050 ± 51 | 2041 | 2217 | 0.12 |
| STA | Stubai Alps | 3454 | 675 | 123 | 355 | 23 | 1.01 ± 0.66 | 23 | 2125 ± 34 | 2118 | 2251 | 0.189 |
| SVA | Southern Valais Alps | 2590 | 193 | 995 | 1412 | 155 | 2.34 ± 2.31 | 66 | 1953 ± 105 | 1927 | 2396 | 0.156 |
| SVR | Sesvenna Range | 3174 | 0 | 120 | 359 | 17 | 0.96 ± 0.43 | 19 | 2275 ± 22 | 2271 | 2370 | 0.141 |
| TVA | Ticino and Verbano Alps | 3272 | 192 | 603 | 898 | 138 | 1.58 ± 1.48 | 87 | 2098 ± 57 | 2093 | 2335 | 0.229 |
| VAP | Varese Prealps | 1648 | 0 | 313 | 410 | 16 | 2.31 ± 2.44 | 7 | 1424 ± 70 | 1417 | 1603 | 0.052 |



| | | | | | | | | | | | |
|---|---|---|---|---|---|---|---|---|---|---|---|
| **VIP** | Vicentine Prealps | 2333 | 34 | 1916 | 2843 | 175 | 2.24 ± 3.66 | 78 | 1917 ± 57 | 1907 | 2135 | 0.091 |
| **WMA** | Weisshorn and Matterhorn Alps | 4470 | 448 | 122 | 424 | 22 | 0.97 ± 0.72 | 23 | 2340 ± 35 | 2336 | 2458 | 0.183 |
| **ZA** | Zillertal Alps | 3499 | 563 | 381 | 863 | 52 | 0.81 ± 0.31 | 64 | 2186 ± 22 | 2186 | 2249 | 0.136 |
| APENNINES | | | | | | | | | | | |
| **AL** | Alburni | 1897 | 0 | 1699 | 2277 | 43 | 3.09 ± 3.2 | 14 | 1643 ± 91 | 1622 | 1886 | 0.025 |
| **APUA** | Apuan Alps | 1937 | 0 | 965 | 1224 | 96 | 2.34 ± 3 | 41 | 1484 ± 111 | 1468 | 1864 | 0.099 |
| **CAAP** | Central Abruzzi Apennines | 1999 | 387 | 559 | 887 | 65 | 2.61 ± 3.44 | 22 | 1746 ± 42 | 1742 | 1862 | 0.117 |
| **GS** | Gran Sasso | 2908 | 90 | 815 | 1795 | 80 | 1.86 ± 1.48 | 43 | 1751 ± 61 | 1738 | 1964 | 0.098 |
| **LA** | Laga | 2457 | 90 | 1066 | 1662 | 132 | 3.15 ± 4.27 | 42 | 1782 ± 48 | 1778 | 1989 | 0.124 |
| **MA** | Matese | 2049 | 60 | 810 | 1211 | 67 | 2.58 ± 2.66 | 26 | 1651 ± 56 | 1641 | 1820 | 0.083 |
| **MJ** | Majella | 2792 | 98 | 766 | 1344 | 65 | 2.25 ± 3.83 | 29 | 2124 ± 106 | 2118 | 2383 | 0.085 |
| **MM** | Monti Marsicani | 2284 | 325 | 507 | 943 | 72 | 1.63 ± 1.77 | 44 | 1941 ± 35 | 1935 | 2066 | 0.142 |
| **MR** | Monti Reatini | 2214 | 369 | 275 | 389 | 47 | 1.63 ± 1.12 | 29 | 1773 ± 38 | 1768 | 1906 | 0.172 |
| **MS** | Monti Sibillini | 2476 | 239 | 409 | 871 | 68 | 1.89 ± 2.11 | 36 | 1692 ± 48 | 1686 | 1902 | 0.166 |
| **MSE** | Monti Simbruini-Ernici | 2155 | 223 | 811 | 1034 | 98 | 2.57 ± 2.28 | 38 | 1818 ± 40 | 1813 | 1977 | 0.12 |
| **PO** | Pollino | 2265 | 0 | 1287 | 3326 | 107 | 6.66 ± 11.52 | 16 | 1910 ± 88 | 1915 | 2147 | 0.083 |
| **SVE** | Sirente Velino | 2484 | 248 | 426 | 1069 | 78 | 1.62 ± 1.56 | 53 | 1904 ± 39 | 1897 | 2101 | 0.182 |
| **TEA** | Tuscan Emilian Apennines | 2163 | 20 | 4387 | 6245 | 345 | 2.85 ± 5.46 | 121 | 1667 ± 45 | 1664 | 1808 | 0.079 |
| **TRA** | Tosco Romagnolo Apennines | 1654 | 47 | 3954 | 5659 | 28 | 1.85 ± 2.01 | 15 | 1301 ± 47 | 1286 | 1427 | 0.007 |
| **WAA** | Western Abruzzi Apennines | 2247 | 35 | 1142 | 1664 | 154 | 2.49 ± 2.62 | 62 | 1844 ± 43 | 1839 | 2045 | 0.135 |

**Appendix B1:**



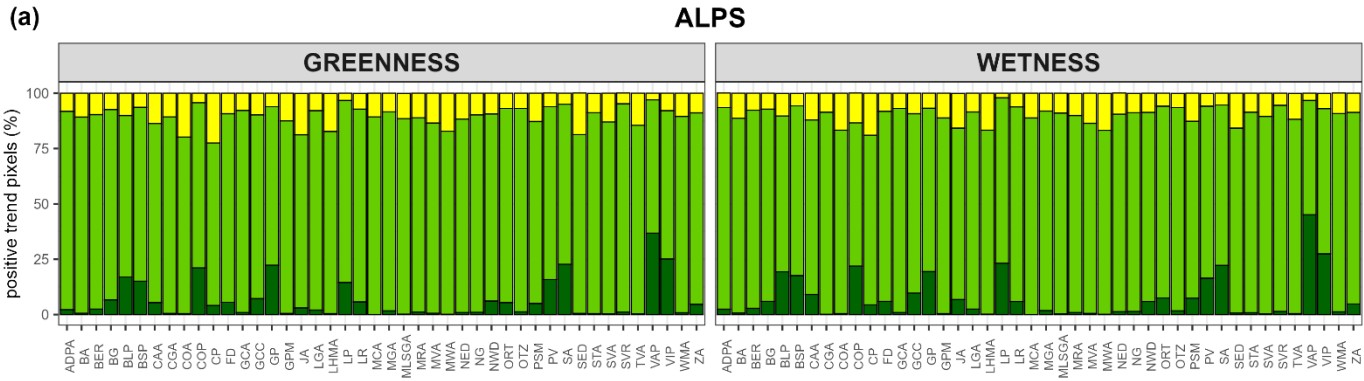

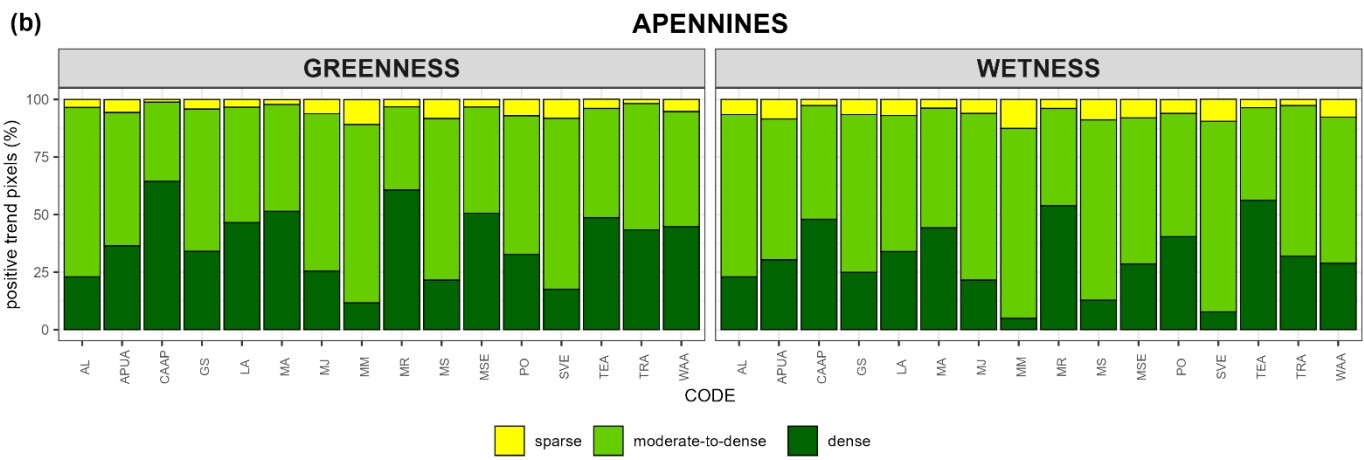

**Figure B1 – Percentage of highly significant positive greenness (left panels) and wetness (right panels) trend pixels of each tree cover density class in the Alps (A) and in the Apennines (B) divided in GMBA mountain groups. Code explainations are in Table A1 (Appendix A).**


## 7. Code availability

Available on request.

## 8. Data availability

Available on request



## 9. Authors contribution

LB: methodology, formal analysis, data curation and writing; DM: methodology, supervision, formal analysis, investigation, writing—review and editing; MG: conceptualization, methodology, investigation, funding acquisition, supervision, writing—review and editing; CU: conceptualization, writing—review and editing; EL: writing—review and editing; RM: writing—review and editing; AV: Conceptualization, supervision, methodology, writing—review and editing.

## 10. Competing interests

Matteo Garbarino is guest editor of the special issue "Treeline ecotones under global change: linking spatial patterns to ecological processes" in Biogeosciences.

## 11. Acknowledgments

This research was cofunded by the European Union – NextGeneration EU, Mission 4, Component 1 (CUP I53D23003180006 - PRIN - 2022 OLYMPUS - Spatio-temporal analysis of Mediterranean treeline patterns: a multiscale approach).

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
