# Peer review of "Forestlines in Italian mountains are shifting upward: detection and monitoring using satellite time-series"

_EGUsphere, 2025_

## Author Comment (AC1)

**REVISION 1 - Anonymous Referee**

Dear revisor, thank you for your valuable revision and insightful comments. We have carefully considered your feedback and tried our best to address each point accordingly.

On behalf of the authors,

Lorena Baglioni

*This manuscript presents an ambitious and methodologically rich study on the upward dynamics of forestlines in Italian mountain ranges, using a 40-year Landsat time series and robust trend detection techniques. The authors propose a semi-automated workflow for delineating the uppermost forestlines and interpreting spectral greenness and wetness trends across canopy classes and topographic gradients.*

*The paper offers valuable insights into large-scale forest recolonisation processes and is grounded in a solid conceptual framework. However, a key aim listed in the abstract is the mapping of the current position of the uppermost Italian forestlines. This core claim is not supported by any form of spatial validation or uncertainty assessment. While statistical comparisons (e.g. Wilcoxon tests, GAMs) are used effectively to interpret spectral trend variability, they do not establish the accuracy or ecological credibility of the detected forestlines themselves. Given the reliance on an automated method to infer a critical ecological boundary, this omission undermines confidence in one of the study's central contributions.*

*The manuscript is generally well written, although there are scattered typographic and formatting issues. Figures would benefit from improved labelling and clarity, particularly in terms of geographical referencing and scalebar readability.*

*I recommend major revisions to address the validation gap and improve clarity for an international audience.*

**Specific Comments**

- ***Forestline validation missing**: A core objective stated in the abstract is to "map the current position of the uppermost Italian forestlines." However, there is no spatial accuracy assessment of these delineated lines. The method's reliability is only indirectly supported through descriptive summaries (e.g. mean elevation), but not through comparison with independent references. I strongly recommend including a validation step - this could involve:*

  o *Visual comparison with high-resolution imagery or historical orthophotos,*

  o *Use of GEDI canopy height profiles,*

  o *Manual digitisation for a subset of sites,*

  o *Field plot data where available.*

  *Including even a partial validation would significantly strengthen the credibility of the forestline mapping method.*

Thank you for your suggestion, but a visual comparison with high-resolution imagery or historical orthophotos and a manual digitisation for a subset of sites would be subjective and time-consuming validations, limiting the replicability and automaticity of the method. Using field plot data is also not applicable to our scale of analysis, as the data collected may not refer to the same year of the Tree Cover Density (TCD) product and may relate to the position of the treeline and not of the closed forest limit (we considered TCD >10%). GEDI canopy height profiles has a worse spatial resolution (30 m) than the TCD (10 m) and therefore it is an unsuitable validating dataset.

We did not include this step in our research because, as we pointed out in the abstract and in the introduction, we aimed to develop an automated methodology for mapping the current position of the uppermost Italian forestlines, replicable in other geographic contexts. For this reason, we decided to use the TCD that is an official and validated large-scale forest cover map.

Referring to the TCD, we added in the manuscript (line 129) "*with a minimum thematic target producer and user accuracies of 90 % (EEA, 2025)*", as reported in the Product User Manual of the Copernicus Land Monitoring Service of the European Environment Agency (EEA).

In addition, we mapped only a subset of the whole forestlines, considering only those at the highest elevation according to a statistical criterion based on the 1st percentile of all the vertical distances. With manual verification or field data, we couldn't validate the relative elevation of each forestline into its GMBA mountain group limit.

For this reason, we underlined the importance of the input data quality on which the automated detection process is based. However, it must be considered that the first aim of the analysis is functional to the spectral analysis that is carried out in a buffer zone.

Nonetheless, thanks to your comment we had the chance to clarify this issue in the manuscript.

- ***Use of Landsat over Sentinel-2*** *(ll. 151–155): The authors should clearly justify the exclusive use of Landsat imagery, given that Sentinel-2 offers higher spatial resolution and comparable temporal coverage (since 2018). If the decision is based on the 40-year archive or data processing/infrastructure concerns, this should be explicitly stated.*

Thank you for your suggestion. As you correctly reported, the decision is based on the longer time span covered by Landsat. At the lines 153 – 158, we modified the text as follow:

"*Sentinel-2 provide images with higher spatial resolution (10 m) but a shorter time span (since 2018) than Landsat. Infact, Landsat images supply multispectral information at medium resolution (30 m pixel size) since 1984 and are commonly used in treeline studies (Arekhi et al., 2018, Bharti et al., 2012, Morley et al., 2019, Garbarino et al., 2024) as they give a good compromise between space and time resolution at regional scale (Hansson et al., 2020). We collected Landsat images acquired from June 1st to September 30th of each year in the period 1984 – 2023, to analyse forest vegetation dynamics during the growing season*"

- ***Line 60***: *Remove placeholder Italian text: "Fare clic o toccare qui per immettere il testo...".*

Amended.

- ***Lines 113–114***: *The transition into peak selection is abrupt. Clarify why peaks were chosen before introducing the selection process, to improve narrative flow.*

We modified the text as follows: "*Being the Italian forestline ecotones the targets of our study, we selected the highest peaks for each mountain group of the Alps and of the Apennines, as*

*defined by the Global Mountain Biodiversity Assessment (GMBA) inventory (Snethlage et al., 2022a, 2022b)."*

- ***Line 115***: *Non-European readers may not be familiar with the Tinitaly DEM. Add a brief description including spatial resolution and accuracy compared to existing sources*

We modified the text following your suggestion and describing that Tinitaly is a 10 m spatial resolution DEM obtained from the harmonisation of each regional Digital Terrain Models.

- ***Line 118***: *Typo — "affetced" → "affected".*

Amended.

- ***Line 119***: *Typo — "alpine bel" → "alpine belt".*

Amended.

- ***Figure 1b***: *Scalebar tick labels are not legible. Improve contrast and font size.*

We modified the figure following your suggestions.

[Figure]

- ***Figure 2***: *No need to highlight the exact location, but indicate whether the example is located in the Alps or Apennines. Add scalebar.*

We modified the caption of the figure 2 specifying "*in the Alps*" and we added the scalebars.

[Figure]

- **Figure 3**: *Typo in caption header — "statistical analysis" should be corrected.*

Amended.

**Technical Corrections**

- **Typographic and formatting issues**: *Address minor typos such as "affetced" (l. 118), "alpine bel" (l. 119) and inconsistent spelling of the indices (e.g. "grenness")*

Amended.

- **Figures**: *All figures should include legible scalebars and clearer geographical context where relevant."*

We improved all figures.

---

## Author Comment (AC2)

**REVISION 2 - Joanna L. Corimanya**

Dear reviewer, thank you for your valuable revision and insightful comments. We have carefully considered your feedback and tried our best to address each point accordingly. On behalf of the authors,

Lorena Baglioni

The manuscript presents a valuable and timely contribution to the study of forestline dynamics in mountain ecosystems under the influence of climate and land use change. The authors propose an automated, reproducible method to detect the uppermost forestlines in the Alps and Apennines, and evaluate long-term trends in greenness and wetness using Landsat time-series and the Contextual Mann-Kendall (CMK) test. The combination of robust spatial datasets, a long temporal window (1984–2023), and a detailed comparison of canopy cover classes adds substantial weight to the conclusions. The literature review is quite thorough, and the flexibility and scalability of the proposed method enhances its utility across global mountain ranges. In addition, the integration of multiple spectral indices (NDVI, EVI, NDMI, TCW, etc.) offers a nuanced perspective on ecological processes such as tree encroachment and canopy densification.

**Major revisions:**

The manuscript would benefit from a careful linguistic edit. There are numerous typographical and grammatical errors (e.g., "threfore", "rispectively", "forestline uo ti 200 m") that impede readability. Sentence structure could be simplified in some sections for clarity and flow.

Thank you for your suggestions. We edited the manuscript according to your useful indications

While the methodological design is sound, a more explicit discussion of the sources of uncertainty (e.g., compositing effects, spatial mismatch between TCD and Landsat, potential overestimation of greening trends due to observation frequency) would improve the robustness of the conclusions.

Here we discussed some points you have mentioned.

- **Compositing effects:**
  At lines 160-162 we added the following text: "*We preferred the medoid compositing technique over traditional compositing approaches because it is more robust to outliers and noise. In fact, it consists of the closest value to the median*".

- **Spatial mismatch between TCD and Landsat:**
  We considered the TCD for i) the forestlines detection and ii) the definition of the canopy cover class of each greenness and wetness trend points during the spectral analysis. In both cases, the spatial mismatch between TCD and Landsat did not affect our analysis. Infact, the detection was independent from the Landsat time series and we analysed spectral trends along a spatial buffer after the definition of the forestlines. In addition, we assigned the canopy cover classes resampling the TCD to 30 m of spatial resolution. A possible spatial mismatch between the Landsat pixel and the modified TCD should not have affected the following analyses as we did not consider the absolute value of canopy cover, but we assigned a class to each point within a range.

At lines 203 -204 we added the following text: "*Because of the large canopy cover classes ranges to which the points were allocated taking into account a mean TCD value, a possible spatial mismatch between the resampled TCD and the Landsat data was ignored*".

- **Potential overestimation of greening trends due to observation frequency:**
  The use of the medoid as a reducer of the values sampled each year during the growing season enable us to minimise overestimation. In fact, the influence of possible outliers along each pixel-based time series was limited and the most representative value among those sampled was considered (as already pointed out in lines 160-162 for the "Compositing effects" discussion). Pixels with missing values for several consecutive years in the time series were not considered and therefore did not affect the trend analysis.
  A risk of overestimation has been assessed in the use of the maximum seasonal NDVI as greening proxy (Bayle et al. 2024), but we did not adopt it.

The authors correctly report that GAMs relating TAU values to elevation and forestline distance were not statistically significant. However, this result could be better contextualized—what does this imply about the spatial consistency or heterogeneity of trends?

The elevation and forestline distance were not statistically significant drivers of greenness and wetness trends for both the Italian mountain ranges. This result is probably due to a combination of topographic, climatic and anthropogenic drivers that must be considered to assess which are the main ones of these spectral trends, considering the relevant environmental differences between the Alps and the Apennines discussed in the study area section.

We added this information in the revised manuscript.

**Minor revisions:**

Line 15: Should say "carried out," instead of, "carried on."

Corrected.

Line 16: I am not sure what is meant by "..and avoided to fix.." Please clarify.

We clarified the sentence. We did not use a predefined minimum elevation threshold for the forestlines detecting method, because other mountain chains can have different altitudinal ranges. For this reason, an established threshold could have affected the replicability in different geographic contexts.

Line 22: There is an extra space between 'respectively' and 'in'.

Corrected.

Line 29: Remove paragraph break.

Removed.

Line 40: Begin a new paragraph at "The ongoing development of remote."

Begun.

Line 43: Remove 'an'.

Removed.

Line 44: Should be 'treeline monitoring' instead of 'treelines monitoring'.

Corrected.

Line 44: The authors should mention the strengths of aerial photography as well, similarly to how they describe the strengths and weaknesses of other methods. For example, aerial photography allows for broader temporal scales compared to satellite-based remote sensing.

We changed the text as follows (lines 42-44): "*At the local scale, aerial photography is commonly used (Ameztegui et al., 2016; Hansson et al., 2020; Nguyen et al., 2024) since it provides older images than satellite ones, although image quality and availability are limiting factors (Morley et al., 2018).*"

Line 57: The clarity of this sentence could be improved with a rewrite. I would suggest changing the sentence to, "..and to study alpine treelines by applying greening proxies like vegetation indices."

We have modified as suggested.

Line 60: A comment appears to be left in by mistake? Remove sentence of a different font color which states, "Fare clic o toccare qui per immettere il testo."

Removed.

Line 68: Spelling error. 'Overcaming' should be 'overcoming'.

Corrected.

Line 75: Change 'photosintetic' to 'photosynthetic'.

Corrected.

Line 78: 'This' should be 'these'.

It refers to "kind".

Line 87: 'Forestlines' should be 'forestline'.

Corrected.

Line 97: Remove paragraph break.

Removed.

Line 98: This sentence should be rewritten for clarity. I suggest, "..while wetness indices are better for detecting gap-filling processes by intercepting the the spectral signal of lower leaf strata."

We have corrected as suggested.

Line 104: Remove one of the periods after 'SE'.

Removed.

Line 105: Should be 'ranges' not 'range'.

Modified.

Line 107: Change 'and/or' to 'and'

Modified.

Line 108: Should be 'ranges' not 'range'.

Modified.

Line 109: Should be 'elevations' not 'elevation'.

Modified.

Line 113: Remove 'A' from, "Alps and Athe Apennines."

Removed.

Line 118: Fix typo in 'affected'.

Corrected.

Line 121: 'Others' should be 'other'.

Amended.

Line 122: Remove 'of'.

Removed.

Lines 133 - 134: Rewrite the sentence for clarity and grammar.

We modified as follows: "We avoided a minimum elevation threshold for the forestlines detection to facilitate the replicability of the method in geographic regions with different altitudinal ranges."

Line 138: Are the points every 10m? If so, authors should increase clarity by stating the points are at 10m intervals instead of 10m distance.

We modified the sentence as suggested.

Figure 1 (c): Yellow is difficult to read in legend. I suggest changing to a different color to indicate the buffer.

We changed it to a dark yellow.

[Figure]

Line 178: Explicitly state the predictor variable(s) that were compared with greenness and wetness to evaluate significance.

The significance (p-value) is not referred to a predictor variable of the detected trends. In the CMK trend test, the p-value of each pixel represents the probability of observing a significant trend in the analysed time series.

Line 195-196: The authors would benefit from increased clarity and specificity in this sentence. Why did they choose two sets of only 40 points? Or is it two sets of 40,000 points, and there is a typographic error? How were the points randomly sampled? Are these different points than the points discussed earlier in the paragraph? Splitting this up into two sentences and providing more detail will greatly improve the readability of the manuscript.

Yes, thank you for pointing out the error. We sampled 40,000 trend points for each mountain range and it is a subset of the previously described points. We modified the sentence as follow: "After this characterisation of the trend points, we randomly sampled two sets of 40,000 points for each mountain range by the "slice_sample()" function of the "dplyr" R package (Wickham et al., 2023). Each set contained an equal number of greenness and wetness significant trend points."

Line 196: 'a' should be 'an'.

Corrected.

Figure 3: This diagram is excellent! It is very thorough and communicates the analysis well.

Thank you, we enjoyed that you appreciated this diagram.

Line 297: There is an extra space before 'Italian'.

Amended.

Line 297: 'emisphere' should be 'hemisphere'.

Amended.

Line 305: 'forestlines' should be 'forestline'.

Lines 311 - 312: Clarity would improve by rewriting the beginning of this sentence. I suggest, "Overall, rising greenness and wetness trends were recorded at both mountain ranges in line with the ongoing natural reforestation processes." The authors should also note that, in my suggested rewrite, I corrected a typo in 'reforestation'.

Thank you for the correction, we modified it.

Line 321: Add a comma after 'general'.

Added.

Line 326: Change 'plant' to 'plants'.

Changed.

Line 327: Change 'associated to' to 'associated with'.

Changed.

Line 331: Remove the comma after 'Apennines'.

Removed.

Line 352: Add a comma after 'Furthermore'.

Added.

Line 363: Change 'climate' to 'climatic'.

Changed.

Line 381: Remove the extra period at the end of the last sentence.

Removed.